# Left Ventricular “Longitudinal Rotation” and Conduction Abnormalities—A New Outlook on Dyssynchrony

**DOI:** 10.3390/jcm12030745

**Published:** 2023-01-17

**Authors:** Ibrahim Marai, Rabea Haddad, Nizar Andria, Wadi Kinany, Yevgeni Hazanov, Bruce M. Kleinberg, Edo Birati, Shemy Carasso

**Affiliations:** 1The Lydia and Carol Kittner, Lea and Benjamin Davidai Division of Cardiovascular Medicine and Surgery, Baruch Padeh Poriya Medical Center, Poriya 1528001, Israel; 2The Azrieli Faculty of Medicine, Bar Ilan University, Zefat 1311502, Israel

**Keywords:** CLBBB, clockwise, longitudinal, rotation

## Abstract

Background: The complete left bundle branch block (CLBBB) results in ventricular dyssynchrony and a reduction in systolic and diastolic efficiency. We noticed a distinct clockwise rotation of the left ventricle (LV) in patients with CLBBB (“longitudinal rotation”). Aim: The aim of this study was to quantify the “longitudinal rotation” of the LV in patients with CLBBB in comparison to patients with normal conduction or complete right bundle branch block (CRBBB). Methods: Sixty consecutive patients with normal QRS, CRBBB, or CLBBB were included. Stored raw data DICOM 2D apical-4 chambers view images cine clips were analyzed using EchoPac plugin version 203 (GE Vingmed Ultrasound AS, Horten, Norway). In EchoPac–Q-Analysis, 2D strain application was selected. Instead of apical view algorithms, the SAX-MV (short axis—mitral valve level) algorithm was selected for analysis. A closed loop endocardial contour was drawn to initiate the analysis. The “posterior” segment (representing the mitral valve) was excluded before finalizing the analysis. Longitudinal rotation direction, peak angle, and time-to-peak rotation were recorded. Results: All patients with CLBBB (*n* = 21) had clockwise longitudinal rotation with mean four chamber peak rotation angle of −3.9 ± 2.4°. This rotation is significantly larger than in patients with normal QRS (−1.4 ± 3°, *p* = 0.005) and CRBBB (0.1 ± 2.2°, *p* = 0.00001). Clockwise rotation was found to be correlated to QRS duration in patients with the non-RBBB pattern. The angle of rotation was not associated with a lower ejection fraction or the presence of regional wall abnormalities. Conclusions: Significant clockwise longitudinal rotation was found in CLBBB patients compared to normal QRS or CRBBB patients using speckle-tracking echocardiography.

## 1. Introduction

Left ventricular contraction efficiency depends in part on synchronous activation of the myocardium. Synchronous activation depends on atrioventricular (AV) synchrony, inter-ventricular (left and right ventricle) synchrony, and intra-ventricular synchrony (within left ventricle). AV synchrony depends on AV conduction, and inter-ventricular synchrony on having intact, fully functioning right and left bundle branches. Intra-ventricular synchrony needs an intact, fully functional left bundle branch and associated subdivisions.

When the conduction system is intact, there is a simultaneous contraction of the myocardium, resulting in efficient systolic and diastolic function. In contrast, conduction delay as in complete left bundle branch block (CLBBB) results in sequential contraction and ventricular dyssynchrony that reduces systolic and diastolic efficiency [1]. Many methods of assessment of this dyssynchrony have been previously suggested at the beginning of the era of cardiac resynchronization therapy (CRT) to pre-assess its presence and extent and optimize results after device implantation [2]. These included use of multiple 2D-Doppler echocardiographic methodologies as m-mode [3], tissue Doppler imaging [4,5,6,7], tissue strain timing [8,9], and tissue strain patterns [10].

Novel strain-imaging techniques using speckle-tracking echocardiography can provide information about regional myocardial mechanics, including not only the timing of myocardial contraction but also patterns of mechanical dysfunction [10]. However, in daily practice when viewing the left ventricle in the apical four chamber view, we noticed a distinct clockwise longitudinal rotation of the left ventricle in patients with CLBBB not addressed by the traditional longitudinal, circumferential, or radial strains that are assessed by speckle-tracking echocardiography. This unique rotation was previously reported by Popovic’ et al. in patients with dilated cardiomyopathy [11].

The aim of this study was to quantify this “longitudinal rotation” of the left ventricle in patients with CLBBB in comparison to patients with normal conduction or complete right bundle branch block (CRBBB) using speckle-tracking echocardiography in an unorthodox way with application of circumferential assessment strain-imaging tools on longitudinal apical long axis views.

## 2. Methods

### 2.1. Patient Selection

This retrospective study included three groups of patients based on the QRS pattern: patients with normal QRS, patients with CLBBB, and patients with CRBBB. We selected consecutive patients for each pattern who had comprehensive echocardiography at the Poriya Medical Center in northern Israel: 19 patients with normal QRS duration and normal ejection fraction, 20 patients with CRBBB and preserved ejection fraction, and 21 patients with CLBBB. Clinical and echocardiographic data were extracted from the institute’s database. The indications for echocardiographic studies were dyspnea and chest pain. QRS pattern and width were determined from ECG tracings that were recorded on the same day of the echocardiographic studies. This study was approved by the local ethical institutional committee.

### 2.2. Echocardiography

Echocardiography (2D-Doppler) was done according to the ASE/EACAVI guidelines [12] on various ultrasound machines (GE Vivid i, E9, E95). Cine clips were stored in full frame rate and not compressed to enable subsequent off-line analysis Ejection fraction, regional wall abnormality (qualitatively analyzed), systolic and diastolic dimensions, and valvular abnormalities were recorded.

### 2.3. Longitudinal Rotation Quantification

The longitudinal strain algorithm is designed to detect myocardial displacement towards the apex. To measure longitudinal rotation, a circumferential strain algorithm is used to assess radial motion relative to a centroid of the ventricle. This method allows one to calculate the direction and extent (peak angle) of the longitudinal rotation [11].

Stored raw data DICOM 2D apical-4 chamber view cine image clips were analyzed using EchoPac plugin version 203 (GE Vingmed Ultrasound AS, Horten, Norway). All analyses were performed offline by a single experienced cardiologist (S.C) specialized in 2D strain imaging. In EchoPac–Q-Analysis, 2D strain application was selected. Instead of using the apical view algorithm for longitudinal strain assessment, the SAX-MV (short axis—mitral valve level) algorithm, used for circumferential strain assessment, was selected for analysis and assessment of “longitudinal rotation”. A closed-loop endocardial contour was drawn to initiate the analysis. After assuring good tracking, the “posterior” segment (representing the mitral valve annulus) was excluded before finalizing the analysis. Longitudinal rotation direction, peak angle, and time-to-peak rotation were recorded.

### 2.4. Statistical Analysis

Categorical variables were expressed as percentages and continuous variables as means ± standard deviations. Pattern characteristics were compared using ANOVA with the Tukey–Kramer post-hoc analysis. Correlation between QRS duration and longitudinal rotation angles were performed using a two-side regression analysis. Chi-square analysis was used for categorical variables. Sensitivity and specificity of longitudinal rotation analysis for the identification of CLBBB was done using ROC (receiver operating characteristic) curve analysis. Statistical significance was defined as a *p* value < 0.05.

Statistical analyses were performed using the MedCalc Statistical Software version 15.6.1 (MedCalc Software bvba, Ostend, Belgium).

## 3. Results

### 3.1. Clinical and Echocardiographic Data

The clinical and echocardiographic data are presented in Table 1. Sixty patients were included in the study: 19 patients with a normal QRS pattern, 21 patients with a CLBBB pattern, and 20 patients with a CRBBB pattern. The mean age of the CLBBB patients (67 ± 12 years) and the CRBBB patients (71 ± 13 years) was significantly higher than the normal QRS patients (61 ± 14 years). Female gender constituted 31.6% of the normal QRS patients, 33.3% of the CLBBB patients, and 25% of the CRBBB patients without significant differences between groups. The QRS duration was shorter among the normal QRS patients with a duration of 95 ± 8 ms, compared to the CLBBB patients (159 ± 15 ms, *p* = 0.00001) and the CRBBB patients (141 ± 11 ms, *p* = 0.0001). On the other hand, the QRS duration of the CRBBB patients was shorter than the CLBBB patients (*p* = 0.0001). The ejection fraction was higher among the normal QRS patients (62 ± 6%) compared to the CRBBB patients (55 ± 14%, *p* = 0.038) and the CLBBB patients (45 ± 17%, *p* = 0.0001). The ejection fraction among the CRBBB patients was higher than the ejection fraction among the CLBBB patients (*p* = 0.043). The left ventricular end diastolic diameter was larger among the CLBBB patients (61 ± 12 mm) compared to the normal QRS patients (53 ± 4 mm) and the CRBBB patients (54 ± 10 mm). The left ventricular end systolic diameter was also larger among the CLBBB patients (47 ± 14 mm) compared to the normal QRS patients (34 ± 5 mm) and the CRBBB patients (38 ± 12 mm). Regional wall motion abnormalities were more common among the CLBBB patients (38.1%) compared to the normal QRS patients (10.5%, *p* = 0.044) and tended to be more common among the CRBBB patients (35%) compared to the normal QRS patients (*p* = 0.07). Congestive heart failure was more frequent among the CLBBB patients (66.7%) compared to the normal QRS patients (5.3%, 0.00006) or the CRBBB patients (30%, *p* = 0.02). On the other hand, congestive heart failure was more common among the CRBBB patients compared to the normal QRS patients (*p* = 0.044).

### 3.2. Longitudinal Rotation

The SAX-MV (short axis—mitral valve level) algorithm used for circumferential strain assessment was easily applied for assessment of “longitudinal rotation” in all participants. Longitudinal rotation angles differed according to conduction patterns (normal QRS, CRBBB, or CLBBB), as demonstrated in Figure 1. In the CLBBB pattern, there was definite and extensive longitudinal clockwise rotation. All patients with CLBBB (*n* = 21) had clockwise rotation with mean four chamber peak rotation angle of −3.9 ± 2.4° degrees (Figure 2).

In a CRBBB pattern, there was definite and extensive longitudinal counterclockwise rotation. However, this rotation pattern was variable among patients with CRBBB. We found 65% of CRBBB patients had counterclockwise rotation, while 35% had clockwise rotation. CRBBB patients had a significantly lower rate of clockwise rotation compared to the CLBBB patients. Overall, patients with CRBBB had minimal mean rotation (0.1 ± 2.2°), which was significantly less than patients with CLBBB (*p* = 0.00001). In patients with a normal QRS pattern (no conduction delay), minimal clockwise rotation was observed. Normal QRS patients had an average rotation of −1.4 ± 3°, which was significantly less than the CLBBB patients (*p* = 0.005) but not significantly different from the CRBBB patients (*p* = 0.093) (Table 1). Clockwise rotation was found among 58% of the normal QRS patients which was not significantly different than among the CRBBB patients (35%, *p* = 0.15) but significantly lower than the CLBBB patients.

Using regression analysis, clockwise rotation was found to be correlated to QRS duration in patients with a non-RBBB pattern (LBBB pattern and normal QRS pattern) (Figure 3). As shown in Figure 3, a longer QRS duration resulted in more clockwise rotation (more negative peak rotation angle). Time-to-peak rotation was not different among the groups: 395 ± 98 ms in normal QRS patients, 339 ± 135 ms in CLBBB patients, and 367 ± 89 ms in CRBBB patients.

Identification of a CLBBB pattern using longitudinal rotation was found to be 95% sensitive and 62% specific for a longitudinal rotation angle smaller or equal to −0.9° (larger negative means more clockwise longitudinal rotation) with an AUC of 0.85 and a *p*-value of <0.001, as seen in ROC curve analysis in Figure 4.

Importantly, the angle of longitudinal rotation was not associated with low ejection fraction or the presence of regional wall abnormality or the echocardiography machine used.

## 4. Discussion

### 4.1. Main Findings

The main findings of this study were: (i) definite and extensive clockwise longitudinal rotation in CLBBB patients, (ii) minimal clockwise or counterclockwise rotation in normal or CRBBB patients, (iii) the angle of clockwise longitudinal rotation correlated with QRS duration in patients with non-RBBB pattern, (iv) the angle of rotation was not associated with low ejection fraction or the presence of regional wall abnormality or the echocardiography machine used.

### 4.2. Longitudinal Rotation

For quantification of longitudinal rotation, we used 2D speckle-tracking echocardiography that has been used for evaluating myocardial mechanics in the last decade as did Popovic’ et al. [11,13]. This method allows for the measurement of left ventricular strain and volume changes throughout the cardiac cycle. Longitudinal, circumferential, and radial strain can be measured easily using dedicated software and appropriate views. Standard strain software assesses rotation only in short axis. Thus, to assess and quantify longitudinal left ventricular rotation and its relation to conduction abnormalities, we applied the short axis strain tool used for circumferential strain on the apical four chamber view used for longitudinal strain. In contrast to Popovic’ et al., we used more than one echocardiography machine, and we excluded the “posterior” segment (representing the mitral valve) before finalizing the analysis.

Our findings confirmed the direction of longitudinal rotation in CLBBB patients was consistent with that expected from daily practice, and its magnitude correlated with the QRS duration (a wider QRS duration in CLBBB patients correlated with more longitudinal clockwise rotation). This finding may explain the mechanical consequences of CLBBB on myocardial function and add to our general base of knowledge on the matter. As it is well known, CLBBB can cause left ventricular dyssynchrony. This dyssynchrony is heterogeneous and depends on the width of the QRS interval and the degree of left ventricular function. Rao et al. [14] reported 72% of heart failure patients with LBBB have documented dyssynchrony on tissue Doppler imaging, which has a heterogeneous regional distribution. Dyssynchrony may be seen in LBBB and normal hearts, but it is does not involve the lateral wall [14]. Contraction patterns in classical CLBBB based on 2D speckle-tracking echocardiographic longitudinal strain analysis (when the septal peak shortening occurred within the initial 70% of the ejection phase, and the lateral wall was initially stretched and had peak shortening after aortic valve closure) were previously reported [15]. This classical contraction pattern has been associated with improved echocardiographic function and survival without the need for heart transplantation or left ventricular assist device implantation in CRT recipients, independent of QRS duration and ischemic etiology [15,16]. Furthermore, a classical CLBBB contraction pattern on longitudinal strain analysis was associated with a better outcome in CRT recipients evaluated by strain software [17]. Recently, Calle et al. proposed a classification of LBBB induced cardiac remodeling based on longitudinal strain patterns [18]. The proposed classification suggests a pathophysiological continuum of LBBB-induced left ventricular remodeling and may be valuable to assess the contribution of LBBB to the degree of left ventricular remodeling and dysfunction. All the mentioned methods rely on timing of activation for evaluation of dyssynchrony and response to CRT and were unable to fully describe mechanical dyssynchrony, the actual target of CRT [10]. Carasso et al. reported left ventricular mechanical strain patterns, rather than just showing the timing of activation, were also found to be highly predictive of the response to CRT [10]. Erikson et al. reported unique flow-specific measures of mechanical dyssynchrony in heart failure patients with LBBB that may serve as an additional tool for considering the risks imposed by conduction abnormalities in heart failure patients and prove to be useful in predicting responses to CRT [19]. However, the last method is based on cardiac MRI, which is not widely used in clinical practice.

A substantial proportion of patients receiving CRT do not improve their functional or echocardiographic status [20,21], regardless of the pre-implantation assessment methods or the post-implantation assessment of resynchronization used. Our method may help in classification of LBBB-induced cardiac remodeling based on longitudinal rotation patterns. This study demonstrated high sensitivity for identification of CLBBB patterns using longitudinal rotation. It may help in predicting the response to CRT by easy identification and quantification of futile myocardial work of longitudinal rotation. Furthermore, this method may help in optimizing CRT after implantation by iterating changes in V-V intervals to minimize the clockwise longitudinal rotation. Clinical studies are needed to validate this method and its additive value over the available clinical, echocardiographic, or device-based optimization techniques [22].

Strain measures myocardial deformation in three dimensions; myocardial shortening from base to apex by longitudinal strain, systolic shortening of the short axis of the ventricle by circumferential strain, and myocardial thickening from endocardium to epicardium by radial strain [23]. One study assessed the usefulness of each type of strain (radial, circumferential, and longitudinal strain) for left ventricular dyssynchrony assessment and its predictive value for a positive response after cardiac CRT. In this study, speckle-tracking radial strain analysis constituted the best method to identify potential responders to CRT [24]. Other studies found combined patterns of longitudinal and radial dyssynchrony could be predictive of left ventricular functional response after CRT [8]. Helm et al. reported dyssynchrony assessed by longitudinal motion was less sensitive to dyssynchrony and followed different time courses than those assessments that utilized circumferential motion [25]. Wang et al. reported the LBBB-contraction pattern identified from radial-strain analysis in the mid-ventricular short-axis view predicted reverse remodeling and outcome following CRT, similar to the longitudinal-strain analysis [26]. Delgado-Montero et al. reported baseline global circumferential strain and global longitudinal strain were significantly associated with long-term outcome after CRT and had additive prognostic value to routine clinical and electrocardiographic selection criteria for CRT [27]. As it seems from the mentioned studies and from other studies, there is no specific type of strain that is the ultimate one that may evaluate dyssynchrony before CRT implantation and to predict the response to CRT. Furthermore, specific types of strains or combinations of strains but not all types of strains were useful in other conditions. For example, Zhang et al. reported global longitudinal but not circumferential or radial strain predicted prolonged hospitalization and the requirement for inotropic support with epinephrine after aortic valve replacement [23].

In this study, we suggest another model of mechanical dyssynchrony longitudinal rotation that can be easily detected by speckle-tracking echocardiography in addition to the well-known longitudinal, circumferential, and radial strains measured by the same technique. In longitudinal rotation, the SAX-MV (short axis—mitral valve level) algorithm instead of apical views algorithms was selected for analysis. The “posterior” segment (representing the mitral valve) had to be excluded before finalizing the analysis. This unique rotation was not found to be associated with low ejection fraction or the presence of regional wall abnormality or the echocardiography machine used. It may be used alone or in combination with other well-known types of strains to better understand the mechanism of dyssynchrony pre- and post-CRT implantation. Our study is in agreement with the study of Popovic’ et al. [11] that showed clockwise longitudinal rotation appeared in the setting of non-ischemic dilated cardiomyopathy with the additional impact of QRS duration. In addition, Popovic’ et al. concluded longitudinal rotation was a moderately strong predictor of end systolic volume decrease during CRT in dilated cardiomyopathy. However, the strain-imaging methodology is still undergoing development, and further clinical trials are needed to determine if clinical decisions based on strain imaging result in better outcome [28,29,30].

### 4.3. Clinical Outlook

The clinical implication of the longitudinal rotation is not known. However, based on our study and the study of Popovic’ et al., this unique and novel method may serve as a simple and reproducible method using speckle-tracking echocardiography with simple modification for finding out optimal candidates for CRT (those who have significant clockwise rotation) and to optimize CRT after implantation by searching for the optimal V-V interval and the optimal left ventricular lead configuration.

### 4.4. Limitations

This is a small study dealing with the longitudinal rotation of left ventricle. The small number of patients limited the statistical power of ROC analysis and regression analysis. The three groups of patients are not homogeneous. Ejection fraction was lower among the CLBBB group compared to the normal QRS group and the CRBBB group. In addition, left ventricular end diastolic diameter and left ventricular end systolic diameter were larger among the CLBBB patients compared to the normal QRS and CRBBB patients. These differences can be partially explained by the fact that CLBBB (compared to normal QRS and CRBBB) can cause inter- and intra-ventricular dyssynchrony that may result in systolic and diastolic dysfunction and increases in cardiac dimensions. In addition, regional wall motion abnormalities were more common among CLBBB patients compared to normal QRS patients and tended to be more common among CRBBB patients compared to normal QRS patients. However, we found the angle of rotation was not associated with low ejection fraction or the presence of regional wall abnormality.

The clinical significance of this method is not addressed in this study. Thus, clinical inferences cannot be derived from this study. Larger studies are needed to validate this method and to investigate if it would help in predicting responses to CRT or help in optimizing CRT post-implantation beyond the well-known methods.

## 5. Conclusions

Significant clockwise longitudinal rotation was found in CLBBB patients compared to normal QRS or CRBBB patients using speckle-tracking echocardiography. The angle of this clockwise longitudinal rotation correlated with QRS duration but was not associated with low ejection fraction or the presence of regional wall abnormality or the echocardiography machine used.

## Figures and Tables

**Figure 1 jcm-12-00745-f001:**
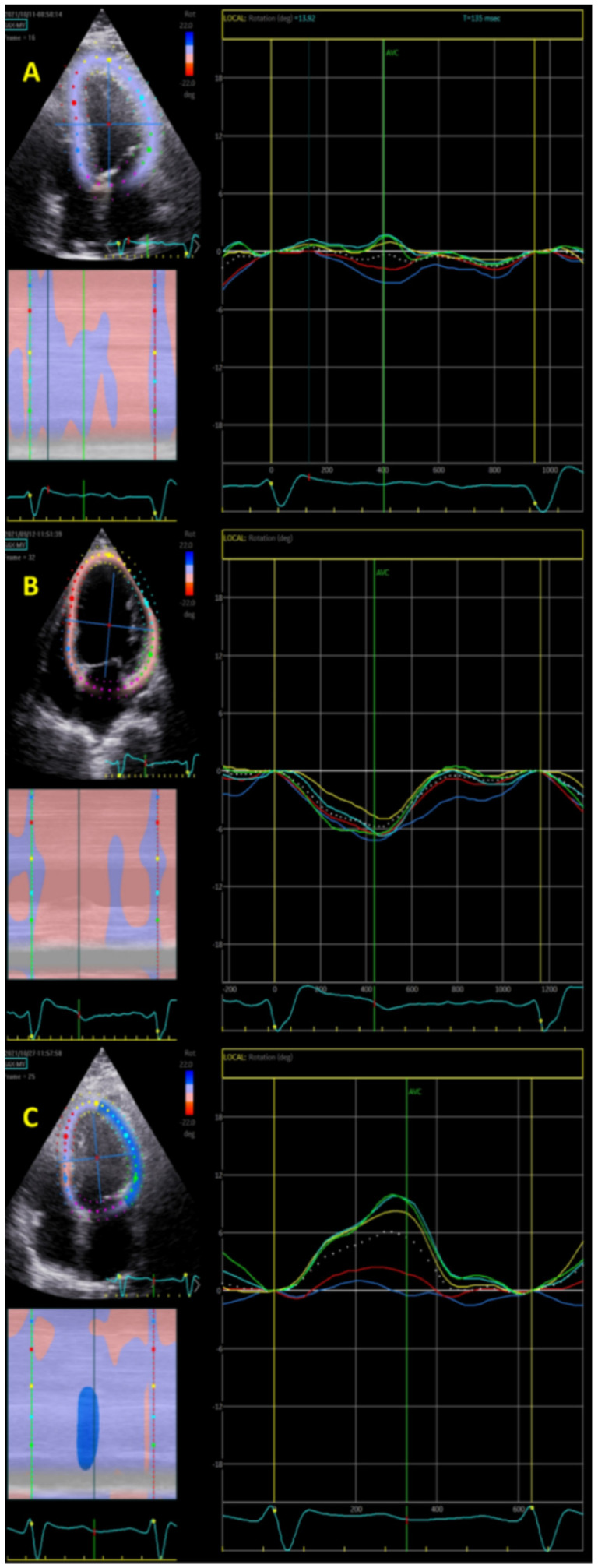
Examples of longitudinal rotation direction and extent in the various patterns analyzed using circumferential rotation algorithm on longitudinal images. Note the “posterior” or annular segment (in pink color) was excluded from analysis. The dotted lines represent the average rotation curve. (**A**)—Normal QRS duration, no conduction delay. Note the minimal clockwise rotation. (**B**)—CLBBB pattern. Definite and extensive longitudinal clockwise rotation. (**C**)—CRBBB pattern. Definite and extensive longitudinal counterclockwise rotation. This rotation pattern was variable among patients with CRBBB.

**Figure 2 jcm-12-00745-f002:**
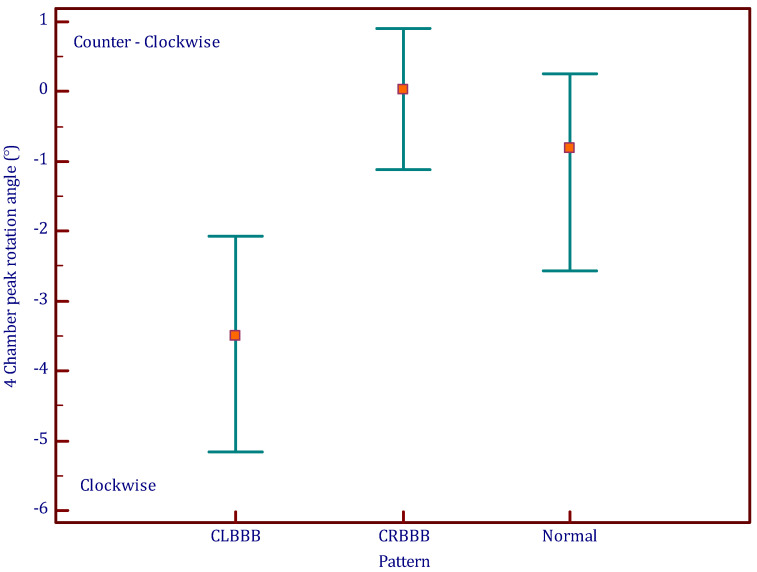
Comparison of longitudinal rotation angle according to conduction pattern. Negative angles are for clockwise rotation, and positive angles are for counterclockwise rotation.

**Figure 3 jcm-12-00745-f003:**
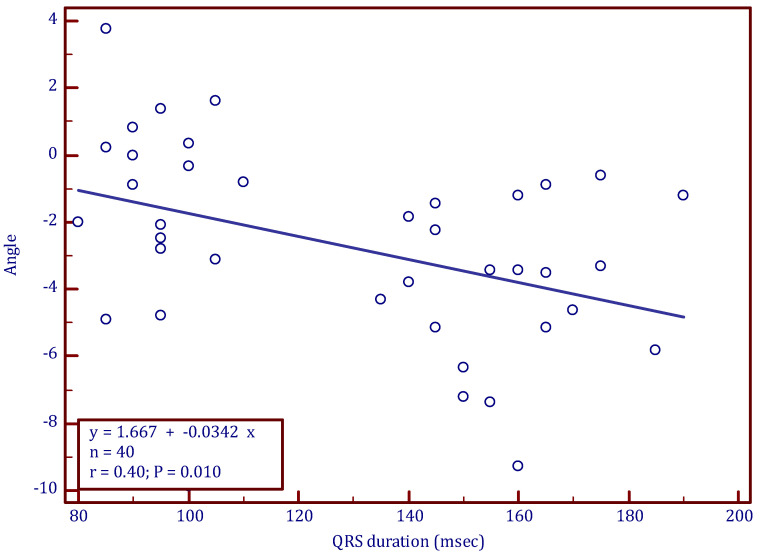
Regression analysis of QRS duration (ms) and longitudinal rotation angle, excluding the CRBBB group.

**Figure 4 jcm-12-00745-f004:**
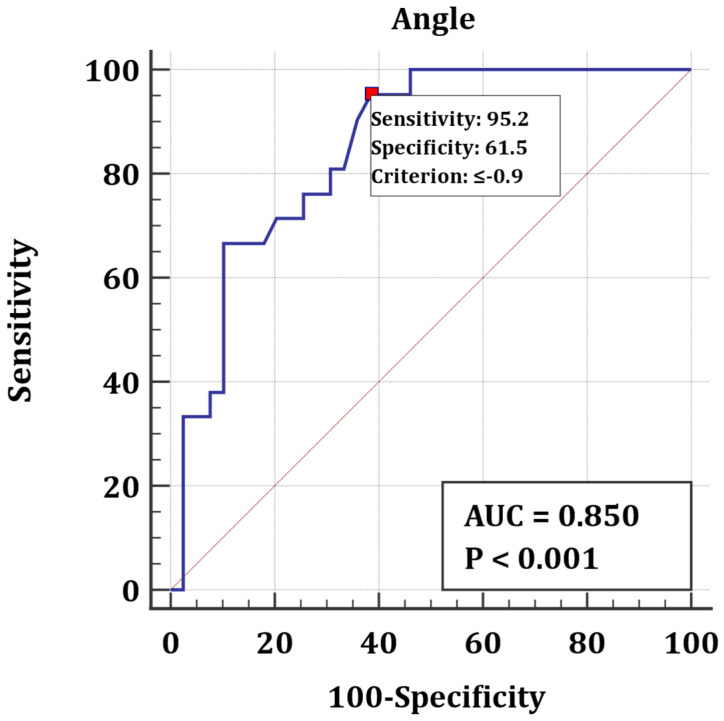
ROC analysis for identification of CLBBB pattern according to rotation angle.

**Table 1 jcm-12-00745-t001:** Clinical and echocardiographic data.

	Normal (*n* = 19)	LBBB (*n* = 21)	*p* Value (LBBB vs. Normal)	RBBB (*n* = 20)	*p* Value (RBBB vs. Normal)	*p* Value (LBBB vs. RBBB)
Age	61 ± 14	67 ± 12	0.0497	71 ± 13	0.008	0.16
Female gender *n*, %	6 (31.6)	7 (33.3)	0.9	5 (25)	0.65	0.56
QRS duration (ms)	95 ± 8	159 ± 15	0.00001	141 ± 11	0.00001	0.00001
Heart rate (min^−1^)	70 ± 12	78 ± 21	0.17	75 ± 13	0.24	0.56
Ejection fraction (%)	62 ± 6	45 ± 17	0.0001	55 ± 14	0.038	0.043
LVEDD (mm)	53 ± 4	61 ± 12	0.004	54 ± 10	0.2	0.04
LVESD (mm)	34 ± 5	47 ± 14	0.0002	38 ± 12	0.06	0.02
Severe valvular disease *n*, %	1 (5.3)	5 (23.8)	0.1	6 (30)	0.1	0.9
Regional WMA *n*, %	2 (10.5)	8 (38.1)	0.044	7 (35)	0.07	0.8
CHF *n*, %	1 (5.3)	14 (66.7)	0.00006	6 (30)	0.044	0.02
Clockwise rotation *n*, %	11 (58)	21 (100)	<0.005	7 (35)	0.15	<0.0005
Longitudinal rotation angle (°)	−1.4 ± 3	−3.9 ± 2.4	0.005	0.1 ± 2.2	0.093	0.00001
Time to peak rotation (ms)	395 ± 98	339 ± 135	0.1447	367 ± 89	0.356	0.437

WMA—wall motion abnormality, CHF—congestive heart failure, LBBB—left bundle branch block, RBBB—right bundle branch block, LVEDD—left ventricular end diastolic diameter, LVESD-left ventricular end systolic diameter.

## Data Availability

Data is unavailable due to privacy or ethical restrictions.

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
