# Peer review of "Left Ventricular “Longitudinal Rotation” and Conduction Abnormalities—A New Outlook on Dyssynchrony"

_jcm, 2023, doi:10.3390/jcm12030745_

Round 1
Reviewer 1 Report
In this study, Marai et al reports and quantifies a clockwise rotation noted in patients with LBBB using echocardiography. One major comment, is that clockwise rotation has been previously described also however this study reports as anecdotal finding. Popovic et al has previously described clockwise rotation in dilated CM PMID: 17664185. This study needs to be referenced and compared against the current analysis.
Minor comments.
1. CLBBB first time use in abstract needs to be expanded/spelled out in full.
2. Abstract conclusion: in in duplicated
3. Methods – 3 groups of – typo
4. Discussion – significant Clockwise – clockwise does not have to be capitalized
5. Has lot of typos and needs to be carefully edited.
Author Response
Thank you for your comments.
- Answer to major comment,: Indeed clockwise longitudinal rotation has been previously described by Popovic et al h in dilated CMP patients ( PMID: 17664185) This study is referenced (ref 11). Our study is in agreement with the study of Popovic´ et al [11] that showed clockwise longitudinal rotation appeared in the setting of non-ischaemic dilated cardiomyopathy, with the additional impact of QRS duration. In contrast to Popovic´ et al, we used more than one echocardiography machine and we excluded the “posterior” segment (representing the mitral valve) before finalizing the analysis. (see text).
- Answer to minor comments: All corrected
Reviewer 2 Report
I thank for the opportunity to review this paper concerning a new echocardiographic way to detect and quantify the grade of myocardial dyssincrony induced by CLBBB.
The authors ingeniously adapt the short axis STI tool to the 4 chamber echo view to determine the myocardial rotation caused by CBBB.
I have several concerns:
-the text requires extensive language reviews. Typos errors are too frequent and really disturb the reading. Many paragraphs use the same word/verb multiple times in consecutive sentences
-the measurements methodology requires a more detailed description: since the authors propose the utilization of an echo tool which was not designed for the 4 chambers view, a deeper description of the rationale of using the short axis tool is needed
-patient selection methodology is poorly described: when the patient were enrolled? Why they needed an echo? ERB number? What does it mean EKG “close” to echo? This doesn’t sound scientific to me
-the clinical, demographic, and echocardiographic results are not enough to understand which population is presented. Since the authors have only 60 patients a better clinical background (healthy or not; etiology of HF?, comorbidities…) and more echo parameters are suggested (LV volumes, valvular competence, standard strain indices…)
-how regional wall motions were assessed? Qualitatively?
-any clinical inference cannot be derived from this study since it is a novel description of an “off label” application of the software for short axis strain which requires validation on larger cohort before suggesting its adoption for clinical purposes
-making ROC analysis and regression analysis on such a limited sample size is misleading and not statistically powered enough
Author Response
Thank you for your valuable comments:
- Answer to comment 1: extensive language review was done.
- Answer to comment 2 (the measurements methodology requires a more detailed description):
The longitudinal strain algorithm is designed to detect myocardial displacement towards the apex. In order to measure longitudinal rotation, a circumferential strain algorithm is used to assess radial motion relative to a centroid of the ventricle. This method allows one to calculate the direction and extent (peak angle) of the longitudinal rotation [11]. In contrast to Popovic et al [11], we excluded the posterior segment (representing the mitral valve).
- Answer to comment 3 (patient selection methodology is poorly described): The patients were retrospectively enrolled for the analysis of this novel method. The indications for ECHO were heart failure symptoms and or chest pain. The indications for echocardiographic studies were dyspnea and chest pain. QRS pattern and width were determined from ECG tracings that were recorded on the same day of the echocardiographic studies. See Methods (patient selection).
- Answer to comments 4 (-the clinical, demographic, and echocardiographic results are not enough ): This is a pilot study aimed to measure the longitudinal rotation in CLBBB in comparison to normal conduction or RBBB unrelated to clinical data. Very important clinical and echocardiographic data are presented in table 1. We are aware that these data are not enough for any clinical inference. We added this limitation to limitation section.
- Answer to comment 5:how regional wall motions were assessed? Qualitatively. we added to methods section.
- Answer to comment 6 (any clinical inference cannot be derived from this study since it is a novel description of an “off label” application of the software for short axis strain which requires validation on larger cohort before suggesting its adoption for clinical purposes:. We agree with the reviewer. The aim of the study was not to investigate the clinical application of this method as we stated in the text. We wrote in the limitation section "The clinical significance of this method is not addressed in this study. Thus, clinical inferences cannot be derived from this study. Larger studies are needed to validate this method and to investigate if it would help in predicting responses to CRT or help in optimizing CRT post implantation beyond the well-known methods".
- Answer to comment 7 (making ROC analysis and regression analysis on such a limited sample size is misleading and not statistically powered enough): We agree with the reviewer. Indeed, this study is small study. . The small number of patients limited the statistical power of ROC analysis and regression analysis (limitation section). However, this pilot study can be used as base for future large studies.
Round 2
Reviewer 2 Report
I thank the authors for their replies to my comments.
My questions have been addressed adequately.
I would suggest a second round of english language editing since I can still find some paragraphs that require some style review:
es: introduction: 1st paragraph: depend has been used 5 times in the sampe paragraph
introduction: 2nd paragraph CTR therapy therapy
discussion: strain measures myocardial deformation in three... Measure has been used 4 times in this paragraph
4.3 Why should we use this method to find candidates for CRT since there is no clinical correlation with EF or other functional echo measures?
I would be more careful in making inferences
Author Response
Thank you for your valuable comments.
- We reviewed the English style and as suggestedh
- Answer to comment "4.3 Why should we use this method to find candidates for CRT since there is no clinical correlation with EF or other functional echo measures?". We agree with the reviewer that clinical implication of this method is not known and we should be more careful in making inferences. However, because extensive clockwise rotation was found in CLBBB and was correlated to QRS duration (not other parameters), we assume that minimizing clockwise rotation by CRT may improve clinical outcome. Popvic et al [11] also suggested that longitudinal rotation may predict success of CRT in DCM subjects. We modified 4.3 section (see details in 4.3 section: clinical outlook section). See also limitations section 4.4: "The clinical significance of this method is not addressed in this study. Thus, clinical inferences cannot be derived from this study. Larger studies are needed to validate this method and to investigate if it would help in predicting responses to CRT or help in optimizing CRT post implantation beyond the well-known methods"